# ACVR1 Function in Health and Disease

**DOI:** 10.3390/cells8111366

**Published:** 2019-10-31

**Authors:** José Antonio Valer, Cristina Sánchez-de-Diego, Carolina Pimenta-Lopes, Jose Luis Rosa, Francesc Ventura

**Affiliations:** Departament de Ciències Fisiològiques, Universitat de Barcelona, IDIBELL, L’Hospitalet de Llobregat, 08907 Barcelona, Spain; j.a.valer@hotmail.com (J.A.V.); csanchezdg@gmail.com (C.S.-d.-D.); carolinapimentacosta@ub.edu (C.P.-L.); joseluisrosa@ub.edu (J.L.R.)

**Keywords:** ACVR1, ALK2, DIPG (diffuse intrinsic pontine glioma), FOP (fibrodysplasia ossificans progressiva), AMH, activin, BMP, bone, heterotopic ossification, cancer

## Abstract

*Activin A receptor type I (ACVR1)* encodes for a bone morphogenetic protein type I receptor of the TGFβ receptor superfamily. It is involved in a wide variety of biological processes, including bone, heart, cartilage, nervous, and reproductive system development and regulation. Moreover, *ACVR1* has been extensively studied for its causal role in fibrodysplasia ossificans progressiva (FOP), a rare genetic disorder characterised by progressive heterotopic ossification. *ACVR1* is linked to different pathologies, including cardiac malformations and alterations in the reproductive system. More recently, *ACVR1* has been experimentally validated as a cancer driver gene in diffuse intrinsic pontine glioma (DIPG), a malignant childhood brainstem glioma, and its function is being studied in other cancer types. Here, we review ACVR1 receptor function and signalling in physiological and pathological processes and its regulation according to cell type and mutational status. Learning from different functions and alterations linked to *ACVR1* is a key step in the development of interdisciplinary research towards the identification of novel treatments for these pathologies.

## 1. ACVR1 Gene and Signalling

The human activin A receptor type I (*ACVR1)* gene (Ensembl: ENSG00000115170), also known as ALK2, is located in chromosome 2q23-q24 [1] and encodes for the 509 amino acid protein (UniProtKB: Q04771). The ACVR1 protein product was initially described as an activin type I receptor [2], and it was found to be expressed in several tissues and different human cell lines [3]. Recent analysis using RNA-sequencing (RNA-Seq) showed that the *ACVR1* gene is ubiquitously expressed in healthy human tissues, with varying expression levels (GTEx Portal) [4]. 

As a member of the BMP/TGFβ receptor family, the ACVR1 protein contains an extracellular N-terminal ligand-binding domain, a transmembrane (TM) domain, an intracellular glycine–serine-rich (GS) domain, and a protein kinase (PK) domain [5,6]. The loop positioned in the helix–loop–helix of the GS domain contains the key residues responsible for ACVR1 activation upon phosphorylation [5]. As a type I receptor, ACVR1 forms heterotetrameric receptor complexes with the type II receptors BMPR2, ACVR2A, and ACVR2B [7]. Such complexes consist of two type I and two type II receptors [8,9]. Upon binding of ligands to the heteromeric complexes, type II receptors transphosphorylate the GS domain of type I receptors. As a result, the kinase domain of type I receptors is activated and subsequently phosphorylates SMAD1/5/8 proteins that transduce the signal [9]. ACVR1 was first described to bind to activin A, a member of the BMP/TGFβ family that usually triggers phosphorylation and activation of SMAD2/3 upon complex formation with type II receptors [2]. Later, ACVR1 was also found to bind several BMPs with distinct affinities, triggering SMAD1/5/8 signalling [10]. Besides canonical SMAD signalling, ACVR1 can activate non-canonical signalling pathways [7,11]. Additional interactions and transduction mechanisms are detailed in each specific section, including activin A signalling via ACVR1(R206H)-ACVR2A/B (detailed in the Fibrodysplasia ossificans progressiva section) and anti-Müllerian hormone (AMH) signalling via ACVR1-AMHR2 (detailed in the Reproductive system section).

## 2. Fibrodysplasia Ossificans Progressiva

### 2.1. FOP Clinical Information

Fibrodysplasia ossificans progressiva (FOP) (OMIM#135100) is a rare genetic autosomal dominant musculoskeletal disease. It is characterised by episodic formation of endochondral heterotopic ossifications (HO) within soft tissues, including fascia, ligaments, tendons, and skeletal muscles [12,13]. The incidence of this rare condition is approximately one patient in 1.6 million, irrespective of sex or ethnical groups [14,15,16]. 

Episodes of heterotopic ossification can be induced by trauma or they can arise spontaneously. The most common initial symptoms of flare-ups include swelling, pain, decreased movement, stiffness, warmth, and redness. FOP flare-ups usually follow specific anatomic patterns, starting in the neck, jaw, shoulders, and back, and advancing to the trunk and limbs. These flare-ups end up in formation of heterotopic endochondral bone. Cumulative ossifications, which usually start at a median age of five years, result in progressive immobilisation, spinal fusion, scoliosis, ankylosed joints, and a reduced life span [17,18,19,20]. The most common cause of death in FOP patients is cardiorespiratory failure from thoracic insufficiency syndrome at a median age of 42 years [21]. Surgery to remove heterotopic ossifications is avoided since it leads to additional inflammation and further bone formation at the surgical site [19]. 

Additionally, some FOP patients present congenital alterations or associated features [13,18]. Congenital malformation of great toes, including hypoplasia or aplasia, is present in most FOP patients [13,18,22]. Congenital malformation of thumbs is also observed in approximately 50% of FOP patients [13,22]. Other variable FOP features include anomalies in the cervical spine (~80% prevalence) [23,24], proximal medial tibial benign osteochondromas (~90% prevalence) [24,25], short broad femoral necks (~70% prevalence) [24], and conductive hearing impairment (~50% prevalence) [22,24,26].

### 2.2. ACVR1 Genetic Mutations in FOP

In 2006, *ACVR1* was identified as the gene responsible for FOP and the mutation c.617G>A, corresponding to Arg206His (R206H) of the *ACVR1* gene, was found commonly linked to this disease [27]. This mutation is localised in the intracellular GS-rich domain of the ACVR1 receptor, and it is present in the majority of FOP patients, usually showing the typical clinical features detailed above [20,24]. The R206H mutation has been extensively analysed in multiple studies, different populations, and geographical groups worldwide, and it has been invariably linked to FOP [24,28,29,30,31,32,33,34,35,36,37,38,39,40]. To date, up to 14 different mutations have been identified to cause FOP, and all of them are localised in the *ACVR1* gene (Figure 1). It has been estimated that the R206H mutation contributes to more than 95% of FOP patients [20,24]. The remaining cases can be attributed to atypical *ACVR1* mutations, including L196P [41,42], P197_F198delinsL [24], R202I [43,44], Q207E [24], F246Y [45], R258S [28,31,33,35,46], R258G [16], G325A [47], G328E [24,34,43,48], G328R [24], G328W [24], G356D [24,33,49], and R375P [24]. 

While multiple case studies and reports have focused on the clinical aspects of R206H carriers, specific clinical features of atypical ACVR1 mutations have also been identified. These include differences among mutations in the age of ossification onset, association with traumatic flare-ups, anatomic patterns of ossification, malformation of toes, alopecia, facial features, hearing impairment, primary amenorrhea, cognitive impairment, severity of the disease, and life expectancy of FOP patients [18,24,52].

### 2.3. ACVR1 Mutants and Altered BMP Signalling

ACVR1 R206H presented mild, ligand-independent BMP signalling, as well as increased responsiveness to BMP stimulation in different cell lines, FOP patient-derived cells, and zebrafish embryonic models. Both alterations contribute to dysregulated BMP signalling [53,54,55,56,57]. In addition to the R206H mutation, almost all *ACVR1* mutations detected in FOP and diffuse intrinsic pontine glioma (DIPG) also presented enhanced responsiveness to BMP, resulting in higher SMAD1/5/8 signalling upon BMP stimulation [50,58,59,60,61]. Interestingly, ACVR1 constructs with a deleted ligand-binding domain and intracellular activating mutations partially retained increased BMP signalling [59]. To explain these observations, it was proposed that changes in signalling arise from alterations in the ACVR1 interaction with FKBP12. FKBP12, a cytoplasmatic FK506-binding protein 1A (*FKBP1A/FKBP12*), was known as a negative regulator of type I receptors. In the absence of the ligand, FKBP12 binds to the GS domain of type I BMP receptors, suppressing their kinase activity and preventing leaky activation of type I BMP receptors. In response to complex formation, phosphorylation of the GS domain by BMP type II receptors releases FKBP12, allowing for complete type I receptor activation [5,6,62]. 

From crystallographic analysis, Chaikuad and colleagues found that multiple *ACVR1* mutations observed in FOP patients could destabilise the inactive state of the receptor, and mutations in the GS domain could impair inhibition by FKBP12 [50]. Moreover, ACVR1 R206H showed impaired binding to FKBP12, resulting in loss of autoinhibition and leaky activation of BMP signalling, potentially contributing to the dysregulated ACVR1 BMP signalling observed [53,54,55,63]. Overexpression of FKBP12 was able to rescue, with different efficacies, the effect of almost all *ACVR1* mutations detected in FOP and DIPG [60]. Remarkably, F246Y mutants showed activation levels equivalent to wild-type ACVR1, while P197_F198delinsL alteration was almost completely resistant to FKBP12 regulation due to site interaction loss [50,60]. Therefore, while most *ACVR1* mutants, including R206H, present impaired binding to FKBP12, some residual interaction is still observed in almost all mutated ACVR1 receptors [60].

### 2.4. Activin A Signalling via ACVR1 Gain of Function Mutants

Activin A, also a member of the TGFβ superfamily, usually signals via the type I receptor ACVR1B and the type II receptor ACVR2A/B through the SMAD2/3 pathway [2,7,64]. Moreover, it was observed that activin A could bind to ACVR1 in the presence of ACVR2A/B. Interestingly, this activin A–ACVR1 binding was not observed in the presence of high levels of BMPR2 [2,61,64]. Given that activin A shares the type II receptor ACVR2A/B with BMP ligands, activin A is able to competitively antagonise SMAD1/5/8 activation upon BMP6/9 stimulation through ACVR2A/B-ACVR1 complexes [65]. Recent breakthrough studies have identified that the ACVR1 R206H mutation confers a neofunction. It was demonstrated that activin A was capable of inducing SMAD1/5/8 activation via ACVR1-R206H [61,66]. Additionally, activin A induced SMAD1/5/8 activation through ACVR1 in other atypical FOP *ACVR1* mutations [59,61]. In response to activin A, this ACVR1-R206H neofunction was able to trigger endochondral ossification in vivo and enhance chondrogenesis of induced mesenchymal stromal cells derived from FOP-iPSCs in vitro [61]. 

It has been demonstrated that activin A signalling in fibro/adipogenic precursors (FAPs) can promote heterotopic ossification in ACVR1-R206H mice models. In these models, activin A is able to induce osteogenic differentiation and promote heterotopic ossification. Moreover, activin A blockade using ActA-mAb prevented HO of transplanted FAPs. Spontaneous and trauma-induced HO was also inhibited or partly reduced in most ActA-mAb treated mice [67]. Interestingly, while heterozygous *ACVR1*-R206H mutations cause FOP, complete loss of the wild-type *ACVR1* allele in *ACVR1*^R206H/+^ mice results in a substantial increase in HO volume [67]. Altogether, this evidence highlights the importance of competition between wild type and mutant ACVR1 receptors in vivo. This competition for ligand binding and shared type II receptors could affect BMP signalling, which highlights the importance of the balance between type I and type II BMP receptor expression levels in the signalling outcome. 

Further emphasising the importance of the relative balance between receptors, several studies have reported that activin A signalling through wild-type ACVR1 is cell type-dependent. Previously, it was assumed that activin A only signals via ACVR1B and ACVR2A/B through SMAD2/3 phosphorylation, and interaction with ACVR1-ACVR2A/B did not stimulate SMAD1/5/8 [66]. However, in immortalised murine embryonic fibroblasts (MEF cells) with wild-type ACVR1, a mild increase in SMAD1/5/8 signalling was observed in response to activin A [59]. These observations have also been reproduced in different myeloma cell lines and HepG2 liver carcinoma cells, describing activin A and activin B signalling via the wild-type ACVR1 receptor through SMAD1/5/8 [64]. Moreover, it was observed that loss of BMPR2 potentiated, and BMPR2 overexpression reduced, activin A signalling via ACVR1-ACVR2A/B through SMAD1/5/8 [64]. Therefore, it was proposed that in cells where ACVR1 is a limiting factor, complex formation preference (either ACVR1-BMPR2, ACVR1-ACVR2A/B, or ACVR1B-ACVR2A/B) determines the signalling outcome, which may be altered in the presence of ACVR1 R206H [64,68]. Cell type-specific signalling was also observed in FOP patient-derived cells. Activin A responses through SMAD1/5/8 were observed in human induced pluripotent stem cells (hiPSCs) from FOP patients. On the contrary, there are reports that hiPSC-derived endothelial cells (iECs) did not show SMAD1/5/8 activation after activin A stimulation, even though both hiPSCs and iECs carried the *ACVR1*-R206H mutation [69]. 

In recent work, a model to integrate the ligand–receptor promiscuity of the BMP pathway and the multiple possible ligand–receptor and receptor–receptor combinations was proposed. All these multiple combinatorial interactions that may vary from cell to cell or within the same cell in different environments defines the final signalling outcome [70]. Therefore, activin A responses through ACVR1 are critically affected by both the mutational status of *ACVR1* and the relative balance of type II, and likely type I, receptors that are able to interact with ACVR1 (Figure 2). 

### 2.5. ACVR1 Non-Canonical BMP Signalling 

In addition to ACVR1-SMAD1 activation, SMAD-independent, non-canonical ACVR1 signalling may also play a role in *ACVR1*-related pathologies (Figure 2). For example, it was observed that lymphocytes derived from FOP patients presented a dysregulated ACVR1-p38 MAPK pathway that could be blocked by p38 inhibitors. However, other inhibitors for non-SMAD signalling pathways, such as ERK or JNK, did not block altered BMP signalling [71]. Additionally, mice lacking *Acvr1* in cartilage presented reduced SMAD responses, but also decreased p38 MAPK activation [72]. 

The PI3K/AKT/mTOR pathway, which is also a non-canonical BMP pathway [11], has been linked to ACVR1. Multiple studies have linked mTOR to trauma-induced HO and FOP. Different inhibitors of mTORC complexes, including rapamycin, have decreased trauma-induced HO [73] and FOP ossification [74,75]. Additionally, the PI3K/AKT/GSK3 pathway has been linked to SMAD1 levels in murine osteoblasts, affecting bone homeostasis and regulation [76]. Recently, PI3K signalling was also linked to HO and FOP since inhibitors of PI3Kα prevented heterotopic ossification in vivo [77]. The Rho/ROCK pathway, a regulator of mechanosensing, cell motility, and cytoskeleton organisation, has also been related to ACVR1. Mutant *Acvr1*^R206H/+^ mice showed increased activation of RhoA, altered cell morphology, and misinterpretation of the tissue microenvironment [11,78].

## 3. Bone Regulation and Skeletal Development 

As a type I BMP receptor, the role of *ACVR1* in skeletal development and homeostasis has been extensively studied [79]. In addition, the relevance of *ACVR1* mutations in FOP has triggered multiple studies on ACVR1 function in HO. Endochondral development of HO in FOP suggests the involvement of ACVR1 not only in osteogenesis but also in inflammation and chondrogenesis. Indeed, different mutant *Acvr1* models have shown enhanced chondrogenesis. It was observed that in vitro overexpression of ACVR1-R206H increased BMP/SMAD signalling and sensitised cells to BMP, increasing osteogenic and chondrogenic differentiation and boosting mineralisation in response to BMP treatment [54]. In addition to BMP dependent responses, BMP independent signalling was also observed for ACVR1-R206H, promoting chondrogenesis [53]. Additionally, *ACVR1*-Q207D viral overexpression in chick forelimb buds promoted cartilage expansion and joint fusion [80]. 

In contrast, *Acvr1* knock-out mice developed cranial and axial skeletal defects and exhibited decreased levels of phosphorylated SMAD1/5 and p38. Therefore, it was proposed that ACVR1 plays coordinated functions with BMPR1A and BMPR1B during endochondral ossification and is required for chondrocyte proliferation and differentiation [72]. Furthermore, inhibition of ACVR1 reduced heterotopic ossification [81]. Knock-down of *Acvr1* also impaired BMP signalling, repressed BMP6-induced osteoblast differentiation, and enhanced myogenic differentiation [82,83]. Surprisingly, conditional disruption of *Acvr1* and *Bmpr1a* in *Osterix*-expressing progenitors after weaning resulted in increased bone mass and altered bone composition. Yet, certain differences were observed regarding the age of induced deletion, and further studies are required to elucidate the functional mechanisms in relation to BMP receptors and bone regulation [84,85]. Furthermore, *ACVR1* is required for periodontium ligament and alveolar bone development, regulation, and remodelling [86]. Loss of *Acvr1* in the dental mesenchyme promoted dental defects. Additionally, *Acvr1* was found to regulate dentin formation, odontoblast cell fate determination in incisors, and odontoblast differentiation in molars [87]. Altogether, these results highlight the essential role of ACVR1 as a regulator of osteogenic and chondrogenic differentiation and its importance in skeletal development. 

## 4. Cardiovascular Development and Cardiac Diseases

BMP signalling has been related to heart morphogenesis and cardiac valve development, and ACVR1 has been studied as an essential receptor in these processes [88,89]. Throughout development, it has been shown that *Acvr1* is required for proper induction of endothelial-to-mesenchymal transdifferentiation (EndMT) during mouse atrioventricular cushion formation [90]. Interestingly, ACVR1 and BMPR1A present non-redundant functions, which are essential in the appropriate arterial pole morphogenesis [91]. Therefore, the deletion of *Acvr1* leads to altered expression of crucial genes for cardiac differentiation and regional identity [91], defects in aortic valve development [92], and promotion of cardiovascular pathologies, including the bicuspid aortic valve [93]. It has recently been described that BMP signalling via ACVR1 is also required for angiotensin II-induced cardiac hypertrophy and fibrosis [94].

ACVR1 has also been related to several cases of congenital heart defects (CHDs). In a screening of human patients with atrioventricular septum (AVS) defects, *ACVR1*-L343P and R307L alleles were found in AVS patients when compared to healthy controls. The R307L mutation, localised in the solvent-exposed region, was not predicted to affect ACVR1 protein structure, and its activity was not significantly affected after experimental validation. On the contrary, ACVR1-L343P, localised in a β-sheet in the PK domain, affected protein structure, displaying impaired kinase activity. These observations were validated in zebrafish embryos, where *ACVR1*-L343P overexpression disrupted atrioventricular canal formation. Altogether, this study highlighted *ACVR1* as a potential causative of AVS defects [95]. 

The mutation H286N, localised in the kinase domain of *ACVR1*, close to the nucleotide-binding site, was reported in a CHD patient with Down syndrome. This mutation showed lower activity compared to the wild-type receptor [96]. Interestingly, while *ACVR1* mutations present in FOP and DIPG usually increase signalling, *ACVR1* mutations in cardiac pathologies, including L343P and H286N, decrease ACVR1 responsiveness (Figure 1). Additionally, ACVR1 was proposed as a candidate for congenital cardiac malformations in a study analysing whole exome sequence data from 342 patients with left-sided lesions [97]. Another proposed candidate was *SMURF1* (SMAD specific E3 ubiquitin protein ligase 1), a ubiquitin ligase that targets BMP pathway SMADs for ubiquitination and degradation [97,98]. Moreover, several cases of ventricular dysfunction have been reported in adult [99] and young [100] patients with FOP, which carry the typical FOP mutation *ACVR1*-R206H [100]. 

## 5. Reproductive System

Many members and receptors of the TGFβ superfamily have been proposed as important regulators of reproductive system development and homeostasis. TGFβs, BMPs, GDFs, AMH/MIS (anti-Müllerian hormone/Müllerian inhibiting substance), activins, and inhibins display different patterns of expression, depending on cellular localisation, stage of maturation, or the developmental stage of the sexual organs. For example, patterns of expression of these ligands or receptors differ between oocyte, granulosa, or theca cells. Patterns of expression and function can also vary during different stages of follicle development [89,101,102]. 

ACVR1 has been substantially associated with reproductive regulation through BMP and AMH signalling. The anti-Müllerian hormone receptor type 2/MIS type II receptor (AMHR2/MISRII) has been previously identified as a type II receptor for AMH/MIS. Different studies have demonstrated that ACVR1 is a type I receptor for AMH/MIS, activating a BMP-like pathway [103,104,105]. BMPR-1A [106] and BMPR-1B [105] have also been reported to be type I receptors for AMH. AMH has been proposed as an important regulator and marker of ovarian function and folliculogenesis. AMH is mainly expressed by follicles between initial follicle recruitment and cyclic recruitment, and its expression decreases after follicle-stimulating hormone (FSH)-dependent follicle growth and selection [107]. AMH inhibits initial follicle recruitment, and it also decreases FSH sensitivity [108]. It has been suggested that the high AMH levels present in polycystic ovary syndrome (PCOS) patients decreases FSH sensitivity, thus disturbing normal follicle selection [107]. Additionally, both BMP6 [109] and BMP7 [110] decrease FSH sensitivity. Considering the previous observations, it has been suggested that alterations in *ACVR1* may include modification of follicle selection, growth, and accumulation in PCOS patients [111]. 

In a cohort of PCOS patients, seven different polymorphisms within non-coding regions of the *ACVR1* gene were studied. Interestingly, some polymorphisms (rs1220134, rs10497189, and rs2033962) were associated with the total follicle number and higher AMH levels, highlighting the importance of *ACVR1* in folliculogenesis in humans [111]. Interestingly, the *ACVR1* polymorphism rs1220134 was also associated with lower breast cancer risk [112]. Furthermore, a possible regulation of ACVR1 levels by cell-secreted exosomes during follicle maturation has been suggested [113]. 

AMH is also responsible for the regression of the müllerian ducts, the primordial anlage of the female reproductive tract, during male sexual differentiation [106]. A FOP patient with the R258G mutation in the *ACVR1* gene that presented gonadal dysgenesis with sex reversal (karyotype 46, XY female) has been reported. It was suggested that the important role of ACVR1 in the mediation of AMH signalling during embryogenesis could be related to the observed sexual alteration [16]. ACVR1, in addition to BMPR1A and BMPR2/ACVR2A, through BMP2 signalling via SMAD1/5/8, contributes to the regulation of pentraxin 3 (PTX3) in human granulosa-lutein cells. PTX3 plays an essential role in fertility, regulating the formation and expansion of the cumulus-oophorus complex (COC), essential for ovulation [114]. Similar downregulation, mediated by ACVR1/BMPR1A of *connexin 43* (*Cx43*) expression, a gap junction protein involved in cell–cell communication and corpus luteum development, was demonstrated in this model. Decreased Cx43 expression by BMP2 reduced cell–cell communication in human granulosa-lutein cells [115].

During embryonic implantation, uterine stromal cells react to embryonic invasion, triggering decidualisation, a process defined by the proliferation and differentiation of these cells. Since BMP2 is essential for embryonic development and ablation of BMP2 in the uterine stroma precludes decidualisation and embryonic development [116], *ACVR1* was studied during uterine peri-implantation [117] and embryonic development [118]. In the absence of *Acvr1*, embryonic invasion and implantation were delayed, and uterine stromal cells failed to undergo the decidualisation process, resulting in sterility. Microarray analysis comparing control and knock-out *Acvr1* murine uteri showed that the ablation of *Acvr1* suppressed *CCAAT/enhancer-binding protein b* (*Cebpb*) expression during decidualisation, which in turn regulates the expression of the progesterone receptor (*Pgr*). This regulation occurred through direct binding between a regulatory sequence in the 3′ UTR of the *Cebpb* gene and SMAD1/5 [117]. Additionally, it was demonstrated that *Acvr1* was essential for proper embryonic development, and embryos lacking *Acvr1* presented defects in primordial germ cell formation [119] and were arrested at late gastrulation, provoking embryonic lethality [118].

## 6. ACVR1 in the Central Nervous System 

Several reports have highlighted the potential importance of ACVR1 in the development and regulation of the central nervous system (CNS). Due to the relevance of ACVR1 in FOP and, more recently, DIPG pathologies, additional evidence has been observed. FOP patients present atypical CNS alterations, including mild cognitive impairment, cerebral cavernous malformations, cerebellar abnormalities, craniopharyngioma, hypoplasia of the brainstem (including pons and cerebellum), and diffuse cerebral dysfunction [16,18,24]. Additionally, neurological symptoms and sensory abnormalities are common in FOP patients [120]. 

BMP signalling plays an important role in CNS development and regulation. For instance, it has been shown that spinal cord development and proliferation, as well as cerebellar granule neuron differentiation, rely on BMP signalling [121]. Postnatal differentiation of cerebellar cells is also modulated by BMP and SMAD1 [122]. Furthermore, BMPs inhibit oligodendroglial differentiation and promote astroglial lineage commitment by neural progenitor cells during the late embryonic and postnatal periods and regulate the balance of adult neurogenesis and differentiation [121,123,124,125,126]. Strikingly, increased BMP signalling diminishes hippocampal neurogenesis and impaired cognition, while decreased BMP responsiveness enhances hippocampal cognition and neurogenesis in mouse models [127]. 

Recent MRI studies have identified FOP patients with T2-hyperintensity of the dentate nucleus regions with dentate nucleus lesions [128,129,130,131]. T2-hyperintense lesions were also observed in frontal periventricular white matter, the spinal cord, and the dorsal pons. Interestingly, BMP signalling increased in response to induced demyelination in the subventricular zone. It was suggested that enhanced BMP signalling in response to demyelination could disrupt reparation since increased BMP responses reduce oligodendrocyte production and increase the number of astrocytes [132]. Additionally, some FOP patients presented either abnormal soft tissue mass surrounding the brainstem or the ventral portion of the pons or lesions in the fourth ventricle, causing hydrocephalus [128,129]. In mouse models with dysregulated BMP signalling, either by BMP4 overexpression or *ACVR1**^R206H^* mutation, demyelinated lesions and focal inflammatory changes of the CNS were observed. Remarkably, demyelinated areas seemed to correlate with the previously detected hyperintense lesions [131]. In a study involving 13 FOP patients, 11 carrying the R206H mutation and two carrying the R258S mutation all displayed thin T2-hyperintense bands at the ventral pons. These brainstem lesions resembled the hamartomatous tissue. All patients showed brainstem dysmorphism, with bulging of the dorsal pons, a thickened pontomedullary junction, and enlarged medulla [129]. Additionally, dentate nuclei abnormalities were observed in all patients. The size of the brainstem lesions did not correlate with age, onset of ossification, or severity of disability. Moreover, brainstem lesions were observed in two FOP patients in the first month of life [129]. These findings possibly suggest either a congenital malformation resulting from impaired regulation of brainstem progenitors during development or early, impaired regulation of the CNS, potentially affected by inflammation and dysregulated BMP signalling due to the *ACVR1* mutations [129,131]. 

All this data correlate mutated *ACVR1* with CNS alterations, suggesting a possible explanation to CNS alterations in FOP patients and a possible link to DIPG pathology. Nevertheless, additional studies should be performed to further explain the alterations exposed in this section. Specifically, we should improve the histological description of the phenotype and confirm the demyelination origin associated to the observed lesions. Additional studies should also demonstrate the relevance of altered BMP signalling in CNS lesions, in the balance of neural populations in FOP and DIPG patients, and the potential transfer of therapeutic strategies between FOP, DIPG, and CNS pathologies.

## 7. BMP Signalling and Pluripotency

BMPs have been shown to play a role in pluripotent stem cell (PSC) regulation. Initial investigations reported that BMP signalling supported self-renewal of mouse embryonic stem cells (mESCs) [133,134], although BMP signalling, in combination with basic fibroblast growth factor, inhibited human ESC self-renewal [135]. In mouse PSCs, BMP signalling induced mesenchymal-to-epithelial transition during the initial stages of reprogramming, promoting early stages of reprogramming of induced PSCs (iPSCs) [136].

Together with the development of human iPSC (hiPSC) protocols, several groups have generated hiPCSs from FOP patients carrying the *ACVR1*-R206H mutation. This increased the array of in vitro models to elucidate the molecular mechanisms and alterations underlying FOP and the study of ACVR1 functions [61,69,137,138,139,140,141]. Originally, it was reported that activation of BMP-SMAD signalling impaired hiPSC self-renewal in FOP patient cells [137]. However, it was also observed that BMP-SMAD signalling promotes reprogramming to pluripotency and hiPSC generation in the early reprogramming phase. Inhibition of BMP-SMAD responses reduced the efficiency of iPSC generation from FOP patients carrying the *ACVR1*-R206H mutation. In contrast, activation of the BMP-SMAD pathway in the early reprogramming phase increased the efficiency of hiPSC generation, while enhancement of BMP signalling in the late reprogramming phase promotes differentiation [141]. 

It is known that *ID* genes, as direct targets of the BMP-SMAD pathway [142], regulate cellular senescence through transcriptional repression of *p16/INK4A* (*CDKN2A*) [143]. p16/INK4A has important roles in iPSC generation since enhanced cell senescence impairs iPSC reprogramming [144]. Therefore, it was found that increased IDs expression, induced by BMP-SMAD activation in response to the *ACVR1*-R206H mutation, promoted reprogramming into iPSCs. Enhanced expression of *ID* genes decreased p16/INK4A protein levels, inhibiting p16/INK4A-mediated cell senescence and bypassing the cell senescence barrier to iPSC reprogramming [141]. Therefore, ACVR1 and the BMP-SMAD-ID pathway play essential roles during reprogramming processes. 

## 8. ACVR1 in Cancer

### 8.1. Diffuse Intrinsic Pontine Glioma

DIPG, a high-grade paediatric brainstem glioma, is a lethal tumour with a mean age of diagnosis of 6–7 years and a median survival of 10 months after detection [145,146,147]. Surgical resection is inadequate due to tumour location within the brain, and radiotherapy is the only temporary treatment available to date [147,148,149]. Molecular knowledge and biological understanding of DIPG has been limited due to intrinsic difficulties regarding biopsy acquisition. As a result, attempts to find an effective therapy have been mainly based on research in glioblastomas from adults, which present highly different molecular alterations compared to DIPG [150]. In 2014, four different groups reported independent genomic analyses of DIPG, which increased our knowledge regarding the genomic landscape of this paediatric glioma. Interestingly, all four groups identified recurrent activating somatic mutations in the *ACVR1* gene [149,151,152,153]. 

Among the molecular markers for DIPG, recurrent mutations in histone variants *H3.3* (*H3F3A*) or *H3.1* (*HIST1H3B/C/I*) with the K27M substitution are the most prevalent alterations present in 70–84% of DIPG cases. *TP53* is the second most common mutation in DIPG patients (42–71%). The third most commonly mutated gene in DIPG is *ACVR1*, with a frequency of 20–32% among DIPG patients [147,149,151,152]. *ACVR1* mutations in DIPG are strongly associated with the presence of the *HIST1H3B* mutation encoding K27M or with activation of the PI3K pathway [149,151,152,153]. Clinically, *ACVR1* mutations are associated with younger age and slightly longer patient survival [149,151]. Surprisingly, Taylor and colleagues observed a strong predominance of females in the DIPG *ACVR1* mutant group compared to the sex distribution in DIPG cases carrying wild-type *ACVR1* [149]. 

Recurrent somatic mutations in the *ACVR1* gene are clustered in the GS-rich domain or in the kinase domain, encoding R206H, Q207E, R258G, G328V, G328E, G328W, and G356D substitutions [149,151,152,153]. Surprisingly, these mutations that had not previously been reported in cancer are the same mutations found in FOP patients, excluding mutation G328V, which is, to date, exclusive of DIPG (Figure 1). These mutations lead to a gain of function of *ACVR1*, increasing BMP signalling, which increases ID proteins. Consistent with this, DIPG samples showed increased levels of signalling through ACVR1 [149,151,152,153]. 

Even though most *ACVR1* mutations are shared between FOP and DIPG, the frequencies of the *ACVR1* mutations are different. In fact, while the *ACVR1*-R206H mutation is found in more than 95% of FOP patients [20,24], *ACVR1* mutations in DIPG are more diversely distributed, comprising R206H (20%), Q207E (2%), R258G (13%), G328V (28%), G328E (24%), G328W (4%), and G356D (9%) (Figure 1) [154]. The high *ACVR1* mutational status in patients with DIPG suggests that ACVR1 directly contributes to DIPG oncogenesis. However, FOP patients with identical ACVR1 mutations do not present increased oncogenic predisposition to cancer or DIPG. 

Brainstem progenitor cells expressing mutant *ACVR1* showed enhanced proliferation and cell survival, and *ACVR1*-R206H was the most potent mutation [150,152]. Additionally, brainstem progenitors expressing *ACVR1*-R206H, with or without *H3.1*-K27M, showed increased markers for EMT and STAT3 signalling, which has been involved in mesenchymal transformation into glioblastomas [150,155]. This information strongly supports the hypothesis of both ACVR1 and H3.1 cooperating in DIPG oncogenesis. Mutant forms of both *ACVR1* and histone *H3.3* K27M increased the levels of phosphorylated SMAD1 and ID protein expression. The co-expression of mutant *ACVR1* and *H3.3* K27M created an additive effect, suggesting that several mechanisms may alter BMP signalling in DIPG [149,152]. 

Very interestingly, mutant *Acvr1* (R206H, G328V, G328E) alone did not cause tumours in mouse models. In combination with *H3.1*-K27M, *p53* loss, and *Pten* loss, mutant *Acvr1* was only able to develop glioma-like lesions. The addition of the PDGFA ligand in the presence of *Acvr1*-R206H and *H3.1*-K27M increased the tumour occurrence. This is consistent with the PDGFRA amplifications detected in human DIPG [151,152]. In addition, a recent study on human DIPG patients to assess spatial and temporal driver mutations pointed to *H3K27M* as the initial oncogenic event in DIPG, which needed a partner driver, such as *ACVR1* (or *TP53*, *PPMID*, or *PIK3R1*) to induce oncogenesis [156]. Altogether, this suggested that mutated *ACVR1* is an oncogenic driver in a context where other driver mutations have occurred. Alterations in PI3K, which has a central role in cancer biology [157], may affect *ACVR1*-related DIPG oncogenesis. As introduced above, activating mutations in PI3K-MAPK pathways have been detected in human DIPGs, including nonsense mutations or deletions in *PTEN* and missense mutations in *PIK3CA* and *PIK3R1* [149,152,156]. 

In addition to the described effects of mutant *ACVR1* in DIPG pathology, further non-confirmed hypotheses have been proposed to explain the link between ACVR1 and DIPG. These include cell migration and apoptosis regulation, altered astroglial cell differentiation, regulation of cancer stem cell self-renewal, or the potential importance of hedgehog proteins and *HOX* genes, which are also affected by BMP signalling [20,158]. All these data have strongly validated *ACVR1* as a cancer driver in DIPG. However, the functional mechanisms of *ACVR1* in the development of DIPG or CNS alterations have not been fully elucidated. 

### 8.2. Adult Glioblastoma

ACVR1 was studied using primary grade IV adult glioblastoma cells to elucidate the importance of BMPs in glioblastoma progression. Overexpression of constitutively active *ACVR1* induced apoptosis in glioma-initiating cells (GICs), while inhibition of ACVR1 reduced apoptosis of GICs [159]. Additionally, mRNA and protein levels of DLX2, a transcription factor involved in development, cell proliferation, and differentiation [160,161], were upregulated in response to either BMP4 or BMP7. Overexpression of *ACVR1* or *DLX2* in primary grade IV glioblastoma cells orthotopically inoculated in nude mice resulted in impeded tumour growth and led to longer survival in vivo [159].

### 8.3. Ovarian Cancer 

The essential functions of ACVR1 in the reproductive system have already been described in this review. Given that AMH, BMPs, and ACVR1 modulate female fertility and ovarian function, ACVR1 has also been studied in ovarian cancer. Wang and colleagues found that stress-induced phosphoprotein 1 (STIP1), a protein adaptor and modulator of heat shock proteins HSP70 and HSP90, was secreted by ovarian cancer cells and found in the blood of ovarian cancer patients [162]. STIP1 was observed to directly interact with ACVR1, activating it and increasing SMAD1/5 phosphorylation downstream of ACVR1. This signal was found to induce proliferation of ovarian cancer cells through this ACVR1-SMAD signalling pathway [163]. Altered BMP signalling in ovarian cancer was also observed in epithelial ovarian cancer cells since autocrine BMP9, which usually signals through ACVRL1, also signals through ACVR1, increasing the phosphorylation of SMAD1/5 and promoting ovarian cancer cell proliferation [164]. 

As explained in the reproductive system section, ACVR1 is one of the BMP type I receptors that binds AMH, which has been associated with ovarian cancer [165]. In granulosa cell tumours (GCTs), sex cord stromal ovarian tumours, it was observed that AMH responses through BMP type I receptors inhibited the growth of GCTs, possibly including the induction of cellular apoptosis [166]. In epithelial ovarian cancer (EOC), the most common type of ovarian cancer, it was suggested that AMH was also able to inhibit proliferation and induce apoptosis of EOC cells, inducing G1/S cell cycle arrest [167]. AMH was also observed to inhibit proliferation and growth of human ovarian cancer cell lines in vitro [168] and in vivo [169]. 

Surprisingly, even though the main signalling pathway SMAD1/5 is shared, BMP or AMH can either promote or inhibit ovarian cancer cell proliferation. AMH inhibits proliferation, while BMP or other ACVR1 activation mechanisms enhance proliferation. These differences may result from different expression abundances of BMP type I receptors in each cellular type [102], different BMP type I receptor affinities to AMH [103], or different environmental conditions altering ACVR1 and/or SMAD downstream signalling. In conclusion, while the main players in ovarian cancer are already known, further in-depth studies are needed to refine the relevance of ACVR1 in ovarian cancer.

### 8.4. Endometrial Cancer

A recent bioinformatic analysis using the mutational data generated by The Cancer Genome Atlas (TCGA) consortium revealed that 3.3% of the endometrial cancer cohort carried *ACVR1* mutations. Interestingly, some of these mutations were observed in FOP and DIPG patients. As in DIPG, *ACVR1* mutations have a tendency to co-occur associated with those on the PI3K-mTOR pathway [170]. 

When analysing mutations annotated in The Catalogue of Somatic Mutations In Cancer (COSMIC) [171], one of the largest curated databases of somatic mutations, which includes additional endometrial cancer datasets [172], *ACVR1* mutations are found in 2.69% (25 mutated samples out of 931 samples tested) of endometrial cancers. Strikingly, 10 out of 25 *ACVR1* mutations found in patients are localised in positions also important in FOP and DIPG. Mutations R202I (identical mutation in FOP) and R375C (R375P is found in FOP) are present in endometrial cancer patients. A small cluster of two patients with G356D (identical mutation in FOP and DIPG) is also observed. The R206H mutation, observed in DIPG, and the most prevalent mutation in FOP, was present in six different patients [171,173].

### 8.5. Prostate Cancer

Prostate cancer has been associated with ACVR1 through endoglin, a TGFβ superfamily auxiliary receptor identified as a marker of tumour angiogenesis and neovascularisation [174]. The canonical signalling mechanism comprises endoglin modulating TGFβ responses (TGF-β1 and TGF-β3) by association to TGF-βR2. This interaction activates TGF-βR1 (either TGFBR1 or ACVRL1) intracellular kinase, phosphorylating SMAD proteins (SMAD2/3 or SMAD1/5, respectively) [174]. However, it has been reported that endoglin also interacts with and activates ACVR1, inhibiting prostate cancer cell migration [175]. On the contrary, ACVR1 could also phosphorylate endoglin, promoting prostate cancer cell migration [176]. This dual role of ACVR1 in prostate cancer cell migration should be further confirmed to ascertain whether this mechanism is also present in different cancer types and *ACVR1*-related pathologies. It could suggest a similar signalling balance depending on the receptor association with endoglin.

### 8.6. Multiple Myeloma

*ACVR1* has been associated with multiple myeloma, a hematological cancer that arises from hematopoietic cells. It was described that BMP9 displayed tumour suppressor activity through ACVR1 signalling, inducing apoptosis and growth arrest in myeloma cells. It was suggested that high levels of endoglin in myeloma cancer cells may scavenge BMP9, impairing the growth inhibitory effects of BMP9 [177]. This hypothesis adds information to the endoglin roles observed in prostate cancer, yet the mechanistic process regulating endoglin-ACVR1 functions should still be defined. 

In recent work, it was observed that activin A and B could signal through wild-type ACVR1 in myeloma cells, activating the SMAD1/5/8 signalling pathway and inducing myeloma cell death. Loss of BMPR2 enhanced activin and BMP induced SMAD1/5/8 signalling, and expression of BMPR2 reduced activin-induced SMAD1/5/8 activity. Therefore, it was proposed that a reduction in BMPR2 levels could increase ACVR1 complex formation with ACVR2A or ACVR2B, thus enhancing ACVR1-mediated SMAD1/5/8 signalling and myeloma cell death [64].

### 8.7. Eye Lens Tumour

*ACVR1* has also been proposed as a regulator of mouse eye fibre cell proliferation and apoptosis [178]. It was suggested that ACVR1 signalling suppresses proliferation in the lens. *P53* knock-out formed epithelioid plaques similar to subcapsular cataracts, while double knock-out *ACVR1/P53* promoted large proliferating tumour-like masses in the lens. Therefore, *ACVR1* was proposed as a potential tumour suppressor for eye lens tumours [179]. 

### 8.8. Erythroleukemia

Overexpression of ACVR1 in the human cell line TF-1, a model of erythroleukemia, resulted in enhanced SMAD1/5 signalling in response to BMP9 and increased cell proliferation. Inhibition of ACVR1 reversed the observed effect, reducing cell proliferation [180].

### 8.9. Head and Neck Squamous Cell Carcinomas 

Head and neck squamous cell carcinomas (HNSCCs) present several risk factors, highlighting smoking and alcohol consumption. Furthermore, several genes with frequent somatic mutations have been associated with HNSCCs [181,182]. Copy number gains of the *ACVR1* gene were reported in HNSCCs in different anatomical regions, including the oral cavity, pharynx, and larynx. Carcinomas with copy number gains of *ACVR1* had increased expression of *ACVR1* and were associated with longer survival [181]. Further research is needed to decipher the molecular mechanism underlying these observations and to define *ACVR1* as a potential HNSCC prognosis marker.

### 8.10. Dual Role of ACVR1 and BMPs in Cancer Biology

Multiple reports have highlighted the apparent dual function of BMPs in cancer [183,184,185]. Different BMPs, or the same BMP ligand in different cancer cell types, can promote or inhibit several cancer hallmarks, including cell proliferation, survival, stemness, and migration. For example, BMP7 was shown to promote, inhibit, or not affect cell growth, depending on the breast cancer cell line [185,186]. Similarly, diverse BMP7 cell type-dependent functions affecting cell growth, invasiveness, and cell survival were observed in different prostate tumour cell lines [187]. BMP2 promoted cell proliferation in ovarian cancer cells [188] and invasiveness in pancreatic cancer cells [189], induced EMT and stemness in breast cancer cells [190], and enhanced proliferation and invasion in nasopharyngeal cancer cell lines [191]. Conversely, BMP2 inhibited growth of human osteosarcoma cells [192]; inhibited colon cancer cell proliferation, migration, and invasiveness [193,194]; suppressed tumour growth of human renal cell carcinoma cell lines [195], and inhibited growth and migration of hepatocellular carcinoma cell lines [196]. 

Multiple additional examples of dual function have been observed and reviewed, raising the question of the role of BMPs as tumour suppressors or oncogenes [183,184]. In light of this evidence, we suggest that the tumour and cell type, together with the tumour microenvironment, are key factors in the regulation of signalling by BMPs. As detailed above, *ACVR1* is a well-known cancer driver due to its essential role in diffuse intrinsic pontine glioma initiation and progression. However, similar to the dual role of BMPs, *ACVR1* could act as a tumour suppressor or oncogene, depending on the cancer type, cell type, or ligand involved (Table 1). Therefore, ACVR1 should be considered a complex regulator of cancer biology. 

## 9. Conclusions and Perspectives

ACVR1 and its signalling were initially studied as a type I BMP receptor. Following the discovery of ACVR1 as the gene responsible for FOP, prior knowledge of *ACVR1* was combined with new findings regarding its function and regulation in order to advance towards a potential treatment for this pathology. Similarly, after the discovery of *ACVR1* as a cancer driver gene in DIPG, the research on this type of cancer was highly benefited from prior accumulated knowledge. Even though *ACVR1* is now mainly known for its relevance in FOP and DIPG, this receptor is also involved in other pathologies. Moreover, due to its relevance in DIPG, *ACVR1* is also being studied in multiple cancers. However, further in-depth studies are needed to refine the molecular basis of ACVR1 function in normal physiology and related pathologies. Therefore, we consider that open, multidisciplinary collaboration and information exchange across research from basic molecular and cellular biology and research from *ACVR1*-linked pathologies, including FOP, DIPG, and other pathologies, will help to promote robust knowledge and to identify novel therapeutic opportunities. 

## Figures and Tables

**Figure 1 cells-08-01366-f001:**
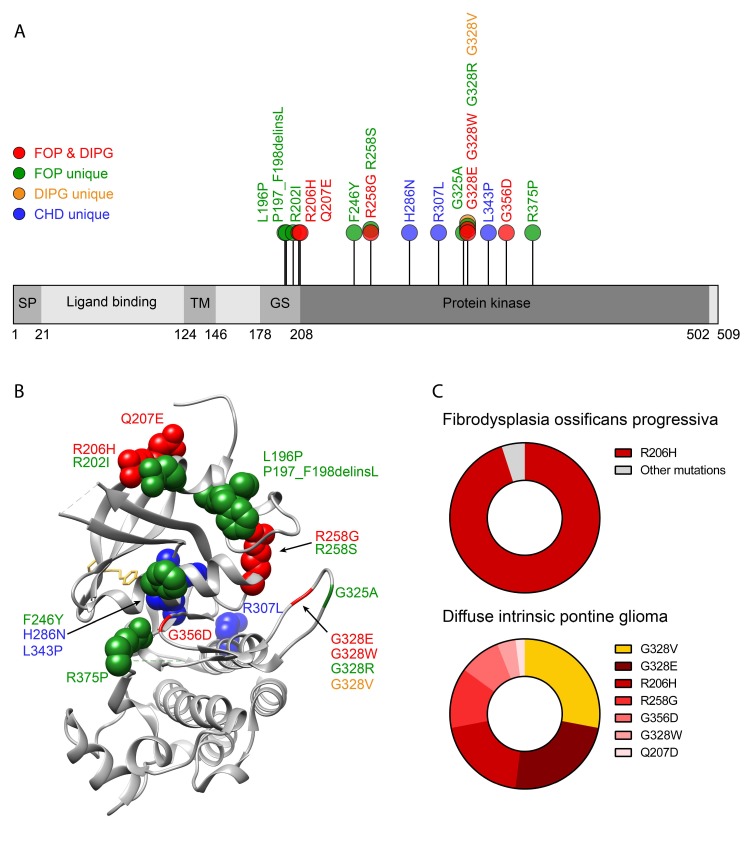
Distribution of recurrent ACVR1 mutations in human pathologies. (**A**) Schematic representation of ACVR1 protein and observed mutations in fibrodysplasia ossificans progressive (FOP), diffuse intrinsic pontine glioma (DIPG), and congenital heart defects (CHD). Domains depicted from amino acids 1 to 509, including signal peptide (SP), ligand-binding domain, transmembrane domain (TM), glycine-serine-rich domain (GS), and protein kinase domain. (**B**) 3D crystal structure of GS and protein kinase domains of ACVR1 protein. Secondary structure elements shown as ribbon diagram. Mutations observed in both FOP and DIPG (red), mutations only observed in FOP (green), DIPG (yellow), or CHD (blue) are labelled in each mutation site. Highlighted amino acids are represented as space-filling spheres. Dorsomorphin, a type I BMP receptor inhibitor, (yellow stick) is represented in the ATP-pocket of ACVR1 kinase. 3H9R structure was obtained from PDB [50] and visualised using UCSF Chimera software [51]. (**C**) Proportion of FOP and DIPG cases affected by ACVR1 mutations.

**Figure 2 cells-08-01366-f002:**
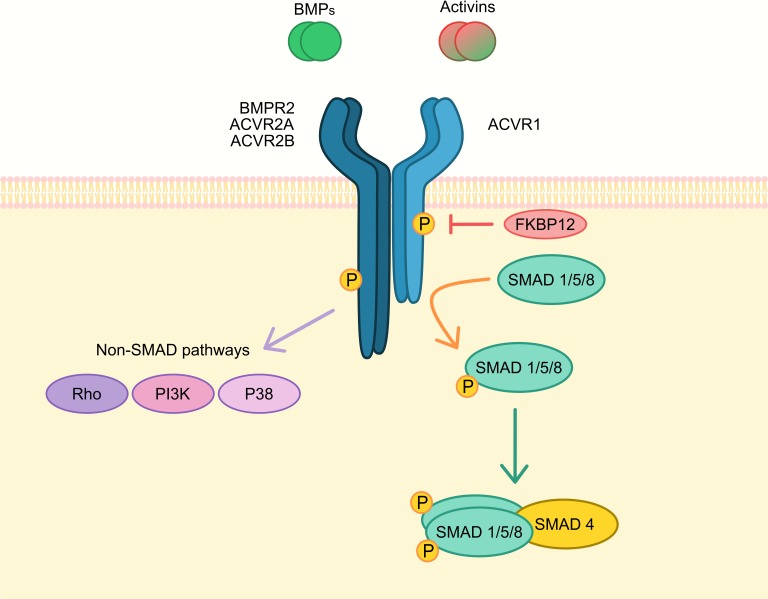
Schematic representation of main ACVR1 signal transduction. ACVR1 is a type I receptor that can interact with type II receptors (BMPR2, ACVR2A, ACVR2B) to form transmembrane heterotetrameric receptor complexes. These complexes can bind ligands including BMPs (e.g., BMP6/7/9) and activins (e.g., activin A/B). Type II receptors can phosphorylate (P) and activate type I receptors, which can trigger canonical SMAD1/5/8 or non-SMAD signal transduction pathways. Phosphorylated SMAD1/5/8 interacts with SMAD4 to translocate into the nucleus and regulate transcription of SMAD target genes. ACVR1 signals through SMAD1/5/8 signalling pathway upon BMP ligand binding. In the presence of wild-type ACVR1 and a high BMPRII/ACVR2A-B expression ratio, activin A could antagonise SMAD1/5/8 signalling through SMAD2/3 and a possible ACVR2A-B competition for ACVR1 or ACVR1B complex formation. In the presence of mutated ACVR1^R206H^, or in the presence of low BMPRII/ACVR1A-B expression ratio in some cellular models, activin A could also signal through SMAD1/5/8.

**Table 1 cells-08-01366-t001:** Role of ACVR1 in different cancer types.

Cancer Type	Function	Hallmark	Role in Cancer	
Diffuse intrinsic pontine glioma	Mutant ACVR1 (including R206H) increases cell proliferation in brainstem progenitors [150,152].	Proliferative signalling	Oncogene	
Mutant forms of ACVR1 lead to increased STAT3 signalling promoting mesenchymal profile and EMT [150].	Epithelial-mesenchymal transition (EMT)	Oncogene	
Glioblastoma	Overexpression of ACVR1 induced apoptosis in glioma-initiating cells and impaired tumour growth [159].	Regulation of programmed cell death	Tumour suppressor	
Suppression of growth
Ovarian cancer	ACVR1-STIP1 interaction promotes human ovarian cancer cell proliferation [163].	Proliferative signalling	Oncogene	
Autocrine BMP9 signalling through ACVR1 promotes ovarian cancer cell proliferation [164].	Proliferative signalling	Oncogene	
AMH signalling reduces granulosa cell tumour growth, increasing apoptosis through type I ACVR1 and BMPR1A/B [166].	Regulation of programmed cell death	Tumour suppressor	
AMH inhibits cell proliferation of epithelial ovarian cancer cells [167].	Proliferative signalling	Tumour suppressor	
AMH inhibits human ovarian cancer cell proliferation [168,169].	Proliferative signalling	Tumour suppressor	
Prostate cancer	ACVR1-Smad1 phosphorylation upon endoglin interaction promotes cell migration [175].	Invasion and metastasis	Tumour suppressor	
ACVR1 phosphorylation of endoglin promotes cell migration [176].	Invasion and metastasis	Oncogene	
Multiple myeloma	BMP9 signalling through ACVR1 induces apoptosis and inhibits cell growth [177].	Proliferative signalling	Tumour suppressor	
Regulation of programmed cell death
Eye lens tumour	ACVR1 signalling inhibits cell proliferation and impairs tumour growth [179].	Proliferative signalling	Tumour suppressor	
Suppression of growth
Erythroleukemia	Overexpression of ACVR1 increases cell proliferation in TF-1 cells [180].	Proliferative signalling	Oncogene

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
