# Peer review of "ACVR1 Function in Health and Disease"

_cells, 2019, doi:10.3390/cells8111366_

Round 1

Reviewer 1 Report

This review article is of very good quality, clear, well written and very useful for scientists working in the field of BMP and TGFb signaling.

I list a few notes:

Page 1, line 11: codifies, I suggest encodes

Page 4, line 141: substitute gain of function with neofunction

Page 5, lines 171-173: the reference citation is correct, however I think that other authors have different findings about activation by activin A in cells of endothelial origin carrying the FOP mutation and activation in different cell types also carrying the FOP mutation. The authors might be more cautious about cell-type specificity of activated signaling in FOP. This is relevant in the study of FOP pathophysiology in which many cell types play a role.

Page 6, line 227: EMT is related to epithelial to mesenchymal transition while endothelial to mesenchymal transition is abbreviated as EndMT

Author Response

We thank the reviewers´s suggestions. We modified the manuscript accordingly to introduce the comments of the reviewer.

Reviewer 2 Report

Valer et al summarize the current understanding of ACVR1 (ALK2) in human physiology and disease. The manuscript is well presented and nicely written. The organization is appropriate and the authors do a great job of addressing the large body of primary data while also making it readable and interesting. In particular, I appreciate the way the authors weave together the complicated signal transduction pathways downstream of ACVR1 and relate this to findings in disease states.

I have no edits or suggestions for the manuscript. Thank you for providing this helpful contribution to the field.

Author Response

We appreciate the positive comments of the reviewer.

Reviewer 3 Report

This is a very good and comprehensive review. I will just add a schematic representation of ACVR1 involvement in cell signalling.  

Author Response

We thank the reviewers´s suggestion. We introduced a new Figure (Figure2) and a graphical abstract showing the involvement of ACVR1 in cell signalling.